# Determining the Photoelectrical Behavior and Photocatalytic Activity of an h-YMnO_3_ New Type of Obelisk-like Perovskite in the Degradation of Malachite Green Dye

**DOI:** 10.3390/molecules28093932

**Published:** 2023-05-06

**Authors:** Miguel Ángel López-Alvarez, Jorge Manuel Silva-Jara, Jazmín Guadalupe Silva-Galindo, Martha Reyes-Becerril, Carlos Arnulfo Velázquez-Carriles, María Esther Macías-Rodríguez, Adriana Macaria Macías-Lamas, Mario Alberto García-Ramírez, Carlos Alberto López de Alba, César Alberto Reynoso-García

**Affiliations:** 1Departamento de Ingeniería Mecánica, Centro Universitario de Ciencias Exactas e Ingenierías, Universidad de Guadalajara, Blvd. Marcelino García Barragán 1421, Guadalajara 44430, Jalisco, Mexico; 2Departamento de Farmacobiología, Centro Universitario de Ciencias Exactas e Ingenierías, Universidad de Guadalajara, Blvd. Marcelino García Barragán 1421, Guadalajara 44430, Jalisco, Mexico; 3Grupo de Inmunología y Vacunología, Centro de Investigaciones Biológicas del Noroeste (CIBNOR), Av. Instituto Politécnico Nacional 195, Playa Palo de Santa Rita Sur, La Paz 23096, Baja California Sur, Mexico; 4Departamento de Ingeniería Biológica, Sintética y de Materiales, Centro Universitario de Tlajomulco (CUTLAJO), Universidad de Guadalajara, Carretera Tlajomulco, Santa Fé, Km 3.5, 595, Tlajomulco de Zúñiga 45641, Jalisco, Mexico; 5Departamento de Ingeniería Electro-Fotónica, Centro Universitario de Ciencias Exactas e Ingenierías (CUCEI), Universidad de Guadalajara, Blvd. Marcelino García Barragán 1421, Guadalajara 44430, Jalisco, Mexico

**Keywords:** YMnO_3_, rare-earth manganite, photodetectors, photocatalysis, cytotoxicity

## Abstract

YMnO_3_ is a P-type semiconductor with a perovskite-type structure (ABO_3_). It presents two crystalline systems: rhombohedral and hexagonal, the latter being the most stable and studied. In the hexagonal system, Mn^3+^ ions are coordinated by five oxygen ions forming a trigonal bipyramid, and the Y^3+^ ions are coordinated by five oxygen ions. This arrangement favors its ferroelectric and ferromagnetic properties, which have been widely studied since 1963. However, applications based on their optical properties have yet to be explored. This work evaluates the photoelectric response and the photocatalytic activity of yttrium manganite in visible spectrum wavelengths. To conduct this, a rod-obelisk-shaped yttrium manganite with a reduced indirect bandgap value of 1.43 eV in its hexagonal phase was synthesized through the precipitation method. The synthesized yttrium manganite was elucidated by solid-state techniques, such as DRX, XPS, and UV-vis. It was non-toxic as shown by the 100% leukocyte viability of mice BALB/c.

## 1. Introduction

In recent years, rare-earth manganites, with the general formula RMnO_3_ (R = Rare Earth cation), have been one of the most studied oxide-type compounds in various fields of science. Most studies are related to their ferroelectric properties [1]; however, other studies have revealed that some of them have a magnetocaloric effect [2], making them a potential candidate to be used as refrigerant materials.

Rare-earth manganites can present two types of crystalline structures. The group of manganites with R = La, Ce, Pr, Nd, Sm, Eu, Gd, Tb, and Dy, presents the orthorhombic-type structure as the most stable. On the other hand, the manganites with R = Y, Ho, Er, Yb, Tm, and Lu mainly present a hexagonal-type structure [3]. Furthermore, it has been observed that manganites with a hexagonal-type structure can adopt the orthorhombic structure at 1000°C and 35–40 kbar of pressure [4].

Of the group of manganites with a hexagonal crystalline structure, the most studied and used is yttrium manganate (YMnO_3_). This manganite has various applications, such as in gas sensors in the detection of H_2_S, CO, and NO_x_ [5,6]; photo-capacitive sensors [7]; and non-volatile memory devices [8]. In photocatalysis, YMnO_3_ has been used with NiO, CeO_2_, and SrTiO_3_ to degrade methyl red dye [9,10,11], while it has been combined with Ag_2_S in the degradation of rhodamine B, methyl orange, and methyl blue dyes [12]. In addition, its photovoltaic effect has been studied for its possible use in solar cells [13].

Regarding the materials used as photodetectors in the visible spectrum, the following stand out: GaS, CdS, GaS, MoS_2_, In_2_O_3_, V_2_O_5_, Cu_2_O, Fe_2_O_3_, NiCo_2_O_4_, GaSe, ZnSe, and Si [14], all with a band gap < 3.10 eV. However, most are complicated to synthesize, and some have a considerable toxicity. On the other hand, the photodetection capacity of YMnO_3_ has already been evaluated in the UV spectrum. YMnO_3_ was deposited on yttrium–zirconium substrates, showing satisfactory results [15].

Considering that most of the studies of yttrium manganate in its hexagonal phase (h-YMnO_3_) are based on ferroelectric and ferromagnetic properties, this work aims to evaluate two applications based on its optical properties, particularly its photodetection capability and photocatalytic activity. Both use wavelengths of the visible spectrum. Additionally, its toxicity is also evaluated. For this analysis, h-YMnO_3_ was synthesized by a simple precipitation method and characterized by XRD, SEM, TEM, XPS, and UV-vis. It should be noted that, to our knowledge, it is the first time that the photodetection capacity at wavelengths of the visible spectrum, toxicity, and the use of formic acid as a key factor for the formation of a new morphology in this oxide have been evaluated.

## 2. Results and Discussion

### 2.1. Characterization

Figure 1 shows the diffraction patterns of the calcined precursor powder at 800, 1000, and 1200 °C. At 800 °C, several diffraction lines appear. Those are identified and associated with the following oxides: Y_2_O_3_ (JCPDS #41-1105), Mn_2_O_3_ (JCPDS #24-0508), and Mn_3_O_4_ (JCPDS #24-0734). Moreover, a few diffraction lines associated with the orthorhombic phase of YMnO_3_ (JCPDS #20-0732) were identified. On the other hand, calcination at 1000 °C produced a few of the main diffraction lines associated with the hexagonal YMnO_3_ phase (JCPDS #25-1079). We should point out that, in this calcination, no diffraction lines were detected for manganese oxide; however, there was a tiny peak at 2θ = 48.54° that is related to Y_2_O_3_. According to the literature [16], remnants of lanthanides oxides have been observed on the diffraction patterns for the hexagonal phases of several rare-earth manganates. However, the contributions of those remnants to photovoltaic applications can be considered irrelevant.

The morphology of the precursor powder before (Figure 2A) and after calcination at 1200 °C (Figure 2B) was analyzed by FESEM. In both cases, a large formation of microrods obelisk-shaped was observed. Before calcination, the thickness of the microrods was around 17.8 μm, while after calcination, it increased to 25.1 μm. Additionally, semi-triangular structures attached to the surface of the microrods appear after calcination. Comparing the morphology obtained in this work with those reported for h-YMnO_3_ [17,18,19], we found that those microstructures have not been reported for this oxide. This suggests that the formic acid reaction with the yttrium and manganese nitrates allows this morphology to form, avoiding the surfactants used.

Figure 3 shows a typical TEM image of the precursor powder calcined at 1200 °C. The image shows spicule-like structures with an approximate thickness of 30.8 µm. The morphology of this image corresponds to the upper part of the obelisks (pyramidon) shown in the SEM micrographs.

We analyzed the chemical composition for h-YMnO_3_ using XPS. The full-spectrum scan (Figure 4A) revealed the presence of the chemical elements Y, Mn, O, and C, whose binding energies agree with those reported in the literature [17,20]. The peak associated with C is due to the carbon-based tape used to fix the sample while analyzed. Figure 4B shows the narrow scan of Y 3d. In this case, two peaks appear identified as Y 3d_5/2_ and Y 3d_3/2_ with a binding energies of 157 and 159.1 eV, respectively. These binding energies are associated with the oxidation state of +3 for Y [21]. Figure 4C shows the narrow scan of Mn 2p, which is characterized by the presence of two peaks centered approximately at 641.3 eV (Mn 2p_3/2_) and 653.1 eV (Mn 2p_1/2_). Furthermore, the Mn 2p spectrum was deconvoluted, from which four Gaussian curves were obtained and identified as Mn^3+^ and Mn^4+^ (dotted lines in Figure 4C,D). The curves centered at 641.6 and 653.3 eV (Mn^3+^) are associated with the +3 manganese oxidation state, while those localized at 644 and 651.7 eV (Mn^4+^) are associated with the +4 [17]. Similarly, the O spectrum 1s was deconvoluted, as Figure 4D shows. Two Gaussian curves centered at 529.1 (O lattice) and 531.2 eV (-OH) were obtained. The area associated with O lattice and -OH was 63.25 and 36.75%, respectively.

The mixed oxidation states in manganese (mainly for +3 and +4) in the h-YMnO_3_ surface were observed [22,23]. The +3 oxidation form in manganese is predominant in rare-earth manganates, while +4 appears due to the oxygen deficiency generated by the synthesis process.

Figure 5A shows the UV-Vis absorbance spectra for the h-YMnO_3_ powder. In this case, two absorption bands appear from 780 to 630 and from 480 to 450 nm. Both are located in the visible wavelength range. They have been reported elsewhere related to the hexagonal yttrium manganate [24]. Moreover, the absorbance and wavelength values recorded in the absorbance spectrum were used to obtain the Tauc plot ((αhν)n vs. *hν*) shown in Figure 5B, where α is the absorption coefficient, *hν* is the photon energy, and n is the electronic transition; for h-YMnO_3_ the value reported for n is ½ [24]. From the Tauc plot, it can be observed that the intersection of the dotted line with the abscissa axis yields a band gap value of 1.43 eV. This value is lower than others reported elsewhere for the oxide hexagonal phase [24,25]. It suggests an apparent spectral response in the visible wavelength.

The conduction band (CB) and valence band (VB) potentials of h-YMnO_3_ were calculated using the following empirical equations [26,27].
(1)$$E_{CB}=\chi -E_{e}-\frac{E_{g}}{2}$$
(2)$$E_{VB}=E_{CB}+E_{g}$$
where *E_CB_* and *E_VB_* represent the associated energy of the CB and VB potentials, respectively; *E_e_* is the energy of free electrons vs. hydrogen (4.5 eV); *E_g_* is the optical band gap; and χ is the electronegativity of the compound. The last parameter was obtained using Equations (3) and (4):(3)$$\chi ={\left\{x_{Y}^{a}\times x_{Mn}^{b}\times x_{O}^{c}\right\}}^{\frac{1}{a+b+c}}$$
(4)$$x_{M}=\frac{E_{af}+E_{I}}{2}$$
where *a*, *b*, and *c* are the numbers of Y, Mn, and O atoms, respectively; and *E_af_* and *E_I_* are the electron affinity and the ionization energy, respectively. Figure 6 shows the graphical results.

### 2.2. Photocurrent Tests

Figure 7 depicts the setup utilized for the photocurrent tests.

Before analyzing the photoelectric material response, the LEDs were characterized. The spectra obtained for each LED showed a monochromatic peak at λ = 405 (violet) (Figure 8A). It should be noted that no contributions from other wavelengths were detected in the spectrum of the LEDs.

Figure 8B shows the photocurrent response for λ = 405 nm. Each on/off cycle lasted 2 min, featuring an irradiance (*E_e_*) of 100 mW/cm^2^. When exposed to this wavelength, the photocurrent increased to 4.55 × 10^−8^ A, while in the dark, the photocurrent decreased until it reached 8.50 × 10^−10^ A. A similar photoelectric response has been found in a P-type oxide semiconductor featuring a perovskite structure [28,29]. It has to be pointed out that the applied voltage also contributes to the photoelectric material response. This is because, even in the dark, photocurrent values, as described elsewhere, were found. It is suggested that the applied voltage allows the mobility of the carrier.

As a result, it is possible to propose a photodetection mechanism as an electron–hole (e^−^/h^+^) carrier function. Those are created once the photons that have an *hν* energy create a shade on the h-YMnO_3_ surface, as described by Equation (5):(5)$$h-YMnO_{3\left(surface\right)}+h\nu_{\left(photon\right)}\to h^{+}+e^{-}$$

The incident photons, which have an *hν* energy, allow the atmospheric oxygen adsorption on the material surface, allowing the electron depletion on the conduction band (Equation (6)).
(6)$$O_{2(gas)}+e^{-}\to O_{2(ads)}^{-}$$

Because h-YMnO_3_ is a P-type semiconductor oxide [5], the electron depletion in the conduction band increases the number of charge carriers in the material (in this case, holes for the valence band), allowing the photocurrent to increase. In contrast, oxygen is desorbed when the LED is off, and the conduction band electrons are recombined with the valence band holes, increasing the electric resistance of the material and causing a decrease in photocurrent. Moreover, further analysis was performed by increasing the irradiance (*E_e_*) from 20 to 100 mW/cm^2^ in dark/light cycles of 2 min, as shown in Figure 9. The photocurrent values show a direct dependency on *E_e_*, since when analyzed from 20 to 100 mW/cm^2^, it shows the photocurrent values of 4.03 × 10^−9^ and 4.55 × 10^−8^ A, respectively.

In order to evaluate the material in other visible wavelengths, the pellet was also exposed to red light with the exact exposition times as before (642 nm red, Figure 10A). Figure 10B shows the photocurrent vs. time graph for an *E_e_* of 100 mW/cm^2^. In this case, the maximum photocurrent value was 6.86 × 10^−9^ A. In addition, the quantitative detection was analyzed in a range of *E_e_* from 10 to 100 mW/cm^2^ (Figure 11). The photocurrent values related to the *E_e_* were 1.66 × 10^−9^ and 6.80 × 10^−9^ A, respectively. The results demonstrate a strong dependency between the wavelength and the photocurrent response. It was observed that, the larger the wavelength, the higher the photocurrent, and the incident photon’s energy is lower.

Due to the Y_2_O_3_ diffraction peak in the h-YMnO_3_ diffractogram, photocurrent measurements were also conducted on a Y_2_O_3_ pellet to evaluate its contribution to the photoelectric response of h-YMnO_3_. Therefore, the pellet was exposed to light with λ = 405 nm under the abovementioned conditions. The graphical results are shown in Figure 12.

As observed, the material exhibits an erratic photoelectric response pattern, in which it is impossible to distinguish the periods of exposure to light. This reveals that the energy *hv* of the photons, at this wavelength, is insufficient to promote the electrons from the valence band of Y_2_O_3_ to its conduction band. Based on these results, it is possible to consider the negligible contribution of the Y_2_O_3_ remnant, detected in diffraction, in the photoelectric response of h-YMnO_3_.

The photocurrent tests show that visible light grants the separation of the carrier on the h-YMnO_3_. Considering this, the photocatalytic activity in the degradation of malachite green dye (MG) using violet light (λ = 405 nm) was evaluated.

### 2.3. Malachite Green Dye Degradation

Malachite green (MG) is a dye classified as class II for health hazards [30], which is carcinogenic. It has been used in the aquaculture industry to avoid economic losses due to parasites, but it is prohibited since one of its metabolites (leucomalachite green, LMG) causes damage in animals, compromising the immune and reproductive systems, and also promotes carcinogenesis [31].

Figure 11A shows the UV-Vis absorbance spectra from 200 to 800 nm of the aliquots of the MG solution containing h-YMnO_3_ after being exposed to *λ* = 405 nm at an *E_e_* = 100 mW/cm^2^. The most significant absorbance peak at 617 nm was used as a reference to obtain the degradation percentage. A 9% degradation occurred in the first 25 min, while after 180 min, the degradation increased to 88.3%. A solution of this dye without oxide was used to evaluate the MG dye degradation due to the violet light exposure. The degradation percentage was determined after 25 and 180 min, with 8 and 19.5% values, respectively (Figure 11B). These results reveal negligible violet light contribution to MG dye degradation. It is worth mentioning that, before and after 120 min of exposure to violet light (*λ* = 405 nm, *E_e_* = 100 mW/cm^2^), the pH of the aqueous solution with the MG dye and the photocatalyst was measured, at 5.2 and 5.8, respectively. Furthermore, the temperature during the photodegradation process remained constant at approximately 25 °C. This is to avoid a further contribution of the temperature in the photocatalytic activity.

According to the results, a relation between the band gap value and the photocatalytic efficiency can be established as follows: the lower the band gap value, the lower the energy required for forming electron–hole pairs and, consequently, the generation of reactive oxygen species is more efficient. Furthermore, based on the photocurrent results, it can be observed that, using the *λ* = 405 nm (3.06 Ev), the generated photocurrent is 6.6 times greater than that obtained with the *λ* = 643 nm (1.43 Ev). These photocurrent results are strongly related to the band gap value of h-YMnO_3_ (1.43 Ev) because the energy associated with λ = 405 nm is 2.2 times greater than the band gap value of yttrium manganite. Thus, the generation of charge carriers is more efficient using violet light.

A kinetic model for the Freundlich degradation model was proposed according to the absorbance values for the UV-Vis spectra (Equation (7)).
(7)$$\frac{C_{0}-C}{C_{0}}=kt^{\alpha }$$
where *C*_0_ and *C* are the same as for Equation (10), t is time, k is the speed constant, and *α* is a chemistry constant. Figure 13C shows the linear fit of the data corresponding to this kinetics model. The correlation coefficient obtained was R^2^ = 0.97. Additionally, other models of degradation kinetics were analyzed, particularly the first-order (R^2^ = 0.95), second-order (R^2^ = 0.92), pseudo-first-order (R^2^ = 0.84), and pseudo-second-order (R^2^ = 0.89). The Freundlich kinetic model for photocatalysis focuses on the dye adsorption/desorption on the photocatalyst surface, and the interaction with the reactive oxygen species (ROS) formed on the material’s surface [32].

Subsequently, the h-YMnO_3_ powder used to degrade the MG dye was recovered and analyzed by XRD (Figure 14). Secondary phases or manganese oxides (Mn_2_O_3_, Mn_3_O_4_, and MnO_2_) in the material bulk were not observed. This implies that, once used as a photocatalyst, the powder does not show any change in its crystalline structure.

Once used as a photocatalyst, the h-YMnO_3_ surface was analyzed using XPS. Figure 15A,B shows Mn 2p and O 1s spectra, respectively. The above spectra were deconvoluted, and the area of the curves associated with the different chemical species was determined. In the case of Mn 2p, the area related to Mn^3+^ and Mn^4+^ was 67.40 and 32.60%, respectively. By contrasting the results obtained before photocatalysis, the associated Mn^4+^ area was increased by 15.22%. This increase can be associated with the interaction between Mn^3+^ ions with the photo-holes produced during the exposure of the material to violet light. In such a way, the valence band’s photo-holes contributed to the oxidation of this metal ion, causing Mn^4+^ to increase (Equation (8)).
(8)$${Mn}^{3+}+h_{VB}^{+}\to {Mn}^{4+}$$

Regarding the spectrum of O 1s (Figure 13B), three Gaussian curves located at 529.2 (O-Lattice), 531.7 (-OH), and 533.7 (C-O) eV were obtained after being deconvoluted. The areas associated with those Gaussian curves were 50.06, 46.15, and 3.79%, respectively. Comparing the area related to −OH with that obtained before being used as a photocatalyst, an increase of 9.4% was quantified. An increment in -OH groups in the oxide photocatalysts, particularly TiO_2_, has been observed after exposing them to light with higher energies than their band gap [33]. The occurrence of a Gaussian curve at 533.7 eV associated with C–O bonds was also noted. Its presence can be attributed to the fragments of the dye molecule (moiety) adsorbed on the surface of the photocatalyst. Thus, the exposure to violet light of h-YMnO_3_ in the MG dye solution promotes dye degradation.

#### 2.3.1. Scavengers Test

In order to evaluate the contribution of the main species involved in the photodegradation of MG dye (particularly, $h^{+},\;{OH}^{•},\;and\;O_{2}^{-}$), disodium ethylenediaminetetraacetic acid (EDTA), isopropyl alcohol (ISPA), and p-benzoquinone (p-BZQ) were used to trap $h^{+},\;{OH}^{•},\;and\;O_{2}^{-}$, respectively. To conduct this test, 0.5 mM of each scavenger was added separately in three 20 mL MG dye solutions, each containing 20 mg of h-YMnO_3_ as a photocatalyst. Figure 16 shows the degradation percentages obtained using each scavenger. As can be observed, the reactive species that had the main contribution to the photodegradation of MG were $h^{+}\;and\;{OH}^{•}$. The $O_{2}^{-}$ radicals had the least influence.

#### 2.3.2. pH Influence

pH is another factor that plays an important role in photocatalysis. Therefore, its influence on the photodegradation of the MG dye was also evaluated. For this purpose, two 20 mL of MG dye (one at pH 10 and the other at 2.2), with 20 mg of h-YMnO_3_, were exposed to violet light (*λ* = 405 nm, *E_e_* = 100 mW/cm^2^). Figure 17 shows the graphic results. It shows that there is a greater efficiency in the photodegradation of the MG dye in a basic environment than in an acid one. This behavior could be attributed to the following: At pH = 10, it is possible that more ${OH}^{•}$ radicals are generated, which correspond to the reactive oxygen species with the most significant contribution to the photocatalytic activity of yttrium manganate. On the other hand, it may be that the acidic environment induces a positive electrostatic charge on the dye and the oxide surface, decreasing the electrostatic attraction between them. Therefore, the adsorption of the GM dye would occur more slowly, causing its photodegradation to occur with less efficiency.

#### 2.3.3. Photodegradation Mechanism of the MG Dye Using h-YMnO_3_ as the Photocatalyst

Considering the results obtained, it is possible to propose the following degradation mechanism based on photo-holes formed in the valence band ($h_{VB}^{+}$) and electrons in the conduction band (Equation (5)). The photo-holes react with −OH/H_2_O, promoting the formation of hydroxyl radicals (OH^•^), as shown in Equations (9) and (10) [11]. These radicals are strong oxidizing agents that contribute to the degradation of the MG dye (Equation (11)).
(9)$$H_{2}O+h_{VB}^{+}\to {OH}^{•}+H^{+}$$
(10)$$-OH+h_{VB}^{+}\to {OH}^{•}$$
(11)$${OH}^{•}+{MG}_{dye}\to Degradation\;products$$

On the other hand, the conduction band electrons also contribute to the formation of another radical, particularly $O_{2}^{-}$, as described by Equation (6). Based on other works using h-YMnO_3_ as a photocatalyst [10,12], this radical also promotes OH^•^ formation, as described in Equations (12) and (13).
(12)$$O_{2}^{-}+e^{-}+2H^{+}\to H_{2}O_{2}$$
(13)$$H_{2}O_{2}+O_{2}^{-}\to -OH+{OH}^{•}+O_{2}$$

Based on the results of the photocurrent tests as well as those obtained using the scavengers, a possible photocatalytic mechanism for the photodegradation of the MG dye using h-YMnO_3_ as the photocatalyst is shown in Figure 18.

### 2.4. Cytotoxic Properties

Figure 19 depicts the cell viability, in which it can be observed that all treatments of h-YMnO_3_ showed the maximum values (ca. 100%) without significant differences (*p* > 0.05), suggesting that it may be a safe material.

## 3. Materials and Methods

### 3.1. Synthesis Process

The precipitation method was used to synthesize h-YMnO_3_. Briefly, 2 mmol of Y(NO_3_)_3_·6H_2_O (99.8%, Sigma-Aldrich, Saint Louis, MO, USA) and 2 mmol of Mn(NO_3_)_2_·4H_2_O (99%, Sigma-Aldrich, Saint Louis, MO, USA) were diluted in 20 mL of formic acid (99.8%, Sigma-Aldrich, Saint Louis, MO, USA), which was used as the precipitant agent. The mixture was stirred for 3 min; as a result, an exothermic reaction was created, allowing abundant gases to be released. Once the process finished, a white precipitate was obtained and moderately stirred for 24 h. The solvents were evaporated in a dry furnace at 120 °C for 24 h, and the brown precursor obtained was calcinated at 800, 100, and 1200 °C.

### 3.2. Characterization

#### 3.2.1. Solid-State Techniques

The analysis of the crystalline structure of the calcined powders was conducted by X-ray diffraction (XRD) using an Empyrean diffractometer (PANalytical, Westborougj, MA, USA) (CuKα1 radiation). On the other hand, the morphology was analyzed using a SEM (TESCAN, MIRA 3 LMU, Brno, Czech Republic). The chemical surface analysis and the oxidation elements states for h-YMnO_3_ were performed with XPS (Thermo Scientific K-α, E. Grinstead, UK), with a monochromatic source Al Kα (1486 eV). In a complementary way, transmission electron microscopy (TEM), using a Jeol JEM1010 microscope (Tokyo, Japan), was used.

#### 3.2.2. Photoelectric and Photocurrent Assessments

The photoelectric characterization was conducted on an h-YMnO_3_ pellet of approximately 2 mm in thickness and 1.2 cm in diameter, which was placed on two AgNi sheets that served as the electrical contacts. The pellet was fabricated using 1 g of the oxide, compressed at 120 kg/cm^2^, and sintered at 500 °C for 6 h. Figure 7 shows a setup diagram for photocurrent testing.

The photocurrent measurements were performed by using a digital multimeter (Keithley DMM 6500, Portland, OR, USA) in the direct mode (DC) by using 5 V. Light-emitting diodes (LEDs), with wavelengths of *λ* = 405 and 642 nm, were used during the photoelectric characterization. Each LED’s irradiance (*E_e_*) was measured using a PM 100D (Thorlabs, NJ, USA), which features a silicon sensor S120VC. The emission spectrum of each LED was obtained using a CCS200 Spectrometer (Thorlabs, NJ, USA). The abovementioned measurements were performed at room temperature (25 °C) in a normal atmosphere.

### 3.3. Photocatalytic Activity

The absorbance spectra for the h-YMnO_3_ powders were obtained using a UV-Vis spectrophotometer (Cary 100, Agilent Technologies, Santa Clara, CA, USA). The photocatalytic activity was evaluated in the malachite green dye (MG) degradation. For this, 10 mg of h-YMnO_3_ were added and dispersed in 20 mL of a malachite green solution at a concentration of 1.5 × 10^−5^ M. The degradation percentage was obtained as described in Equation (14):(14)$$\%\;Degradation=\frac{A_{0}-A}{A_{0}}*100=\frac{C_{0}-C}{C_{0}}*100$$
where *A*_0_ and *A* correspond to the absorbance values of the solution with the dye before and after exposure to light, respectively. Similarly, *C*_0_ and *C* represent the concentration of the solution before and after being irradiated with light, respectively. The absorbance spectra measurements were performed in a NanoDrop 2000 spectrophotometer (Thermo Scientific, Waltham, MA, USA).

### 3.4. Cytotoxicity Assay

#### 3.4.1. Protocol to Obtain Leukocytes

The cells used came from the spleen of six healthy male BALB/c mice of approximately 2 to 3 weeks of age and whose weight was 21 ± 2 g, which were sacrificed with the prior approval and guidance by the ethical committee for animals from the Biotherium of Centro de Investigaciones Biológicas del Noroeste (CIBNOR), La Paz, Mexico. A concentration of 1.1 × 10^6^ cells/mL was standardized in an RPMI-1640 medium (GIBCO, Thermo Scientific, Massachusetts, USA) plus 10% fetal bovine serum (GIBCO, Thermo Scientific, Massachusetts, USA). Subsequently, in 24-well flat-bottom plates, 900 µL of the splenocyte suspension was added to each well and then stimulated with 100 µL of h-YMnO_3_ suspended in water at concentrations between 50 and 200 µL/mL, which were sonicated for 5 to 10 min. Finally, the cells were incubated for 24 h at 37 °C (85% RH, 5% CO_2_).

#### 3.4.2. Cell Viability Test (Alamar Blue Assay)

The assay followed the protocol of Riss et al. [34], with some modifications. Briefly, splenocytes were transferred into 96-well plates (1.1 × 10^6^ cells/mL), and 10 µL resazurin solution (Sigma, St. Luis, MO, USA) was added to each well. The plate was then incubated (37 °C, 24 h, 85 HR, and 5% CO_2_). To quantify the viability after the incubation period, a fluorescence plate reader was used at 530 nm excitation and 590 nm emission (Varioskan™ Flash Multimode Reader, Thermo Scientific). A 10% (*v*/*v*) DMSO solution plus the cells was used as the cytotoxic control, and water plus the cells as the positive control. The percentage of cell viability was calculated as described in Equation (15):(15)$$\%\;of\;cell\;viability=\frac{F_{mat}-F_{blank}}{F_{control}-F_{blank}}*100$$
where *F_mat_* is the fluorescence of the cells in the presence of material, *F_blank_* is the fluorescence of wells in the absence of the cells, and *F_control_* is the fluorescence of cells without the material.

#### 3.4.3. Statistical Analysis

The experiments were performed three times, and the results were expressed as the mean (±SD) using GraphPad Prism 8.0.1 software using a one-way ANOVA with Dunnett’s post-test. Statistical significance was established at *, **, or ***, indicating *p* < 0.05, *p* < 0.001, or *p* < 0.0001, respectively.

## 4. Conclusions

In the present work, the photoelectric response and the photocatalytic activity of the hexagonal phase of YMnO_3_ were evaluated. Both use wavelengths of the visible spectrum. Additionally, the toxicity in mouse leukocytes was also assessed. To obtain it, this oxide (with a perovskite-type structure) was synthesized using a simple precipitation method, obtaining a band gap of 1.43 Ev (lower than many associated with this oxide). The diffraction patterns revealed the hexagonal phase at 1200 °C; however, a Y_2_O_3_ remnant was observed, which did not contribute to the photoelectric response. SEM micrographs showed the abundant formation of obelisk-like microstructures, which have not been reported for this oxide. The TEM images agreed with the morphology observed by SEM. Before and after photocatalysis, XPS spectra revealed the presence of mixed oxidation states for manganese (Mn^+3^, Mn^+4^). The +3 oxidation state was the predominant one; however, increases in the +4 oxidation state, as well as -OH groups, were observed after photocatalysis. Regarding the analysis of its photoelectric behavior, the wavelength and optical irradiance were crucial factors that contributed to the photoelectric response. In such a way, the photocurrent decreases when the wavelength is increased, and vice versa. However, when the optical irradiance increased, the photocurrent also increased. Regarding its photocatalytic activity, the degradation of more than 90% of the MG dye in 180 min was obtained, with the pH and the band gap being determining factors in its photocatalytic efficiency. The main chemical species involved in photodegradation were photo-holes and OH radicals. On the other hand, it should be noted that this is the first time that the toxicity of this material in mouse leukocytes has been evaluated. However, even if this material proved safe for mouse leukocytes (since cell viability remained 100%), assessing their impact in an in vivo model is essential.

## Figures and Tables

**Figure 1 molecules-28-03932-f001:**
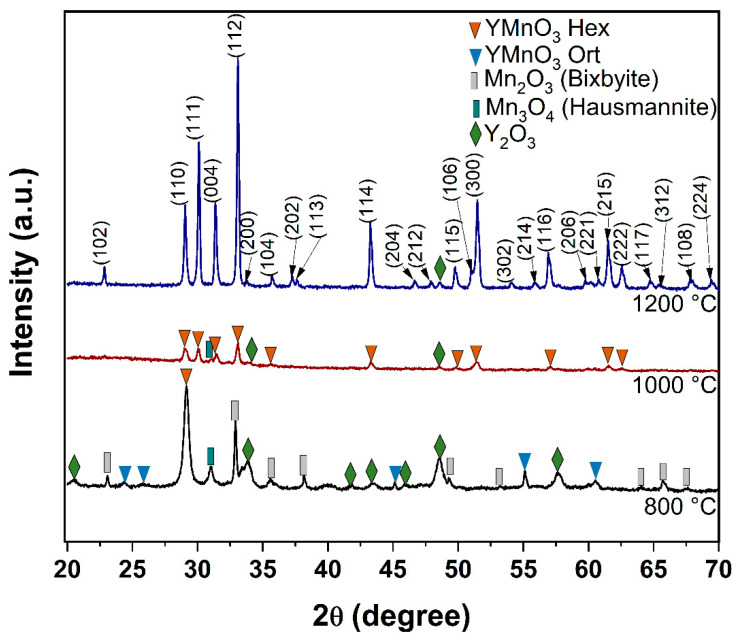
Diffraction patterns of the precursor powder calcined at 800 °C, 1000 °C, and 1200 °C.

**Figure 2 molecules-28-03932-f002:**
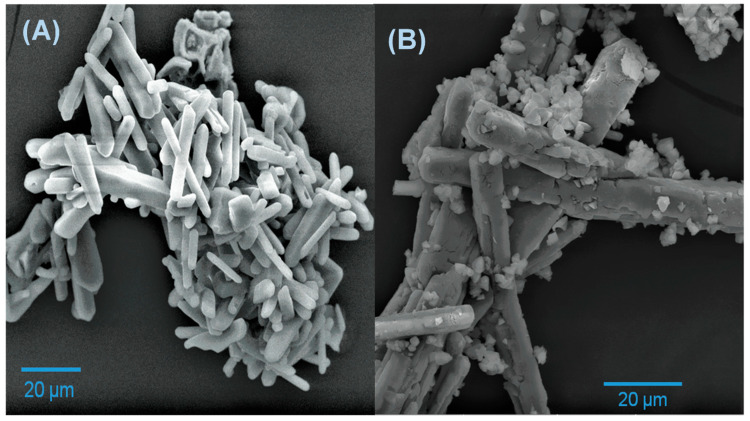
SEM images of the (**A**) precursor powder and (**B**) calcined powder at 1200 °C.

**Figure 3 molecules-28-03932-f003:**
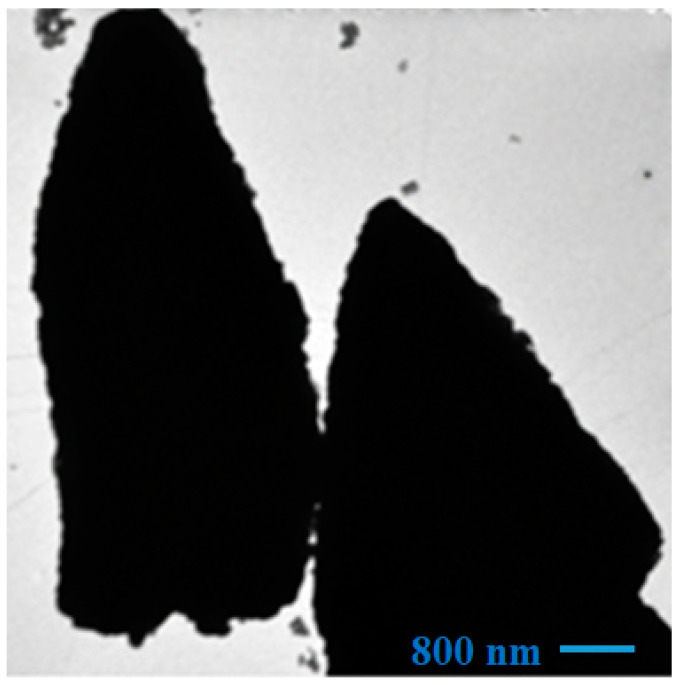
TEM images of the precursor powder calcined at 1200 °C.

**Figure 4 molecules-28-03932-f004:**
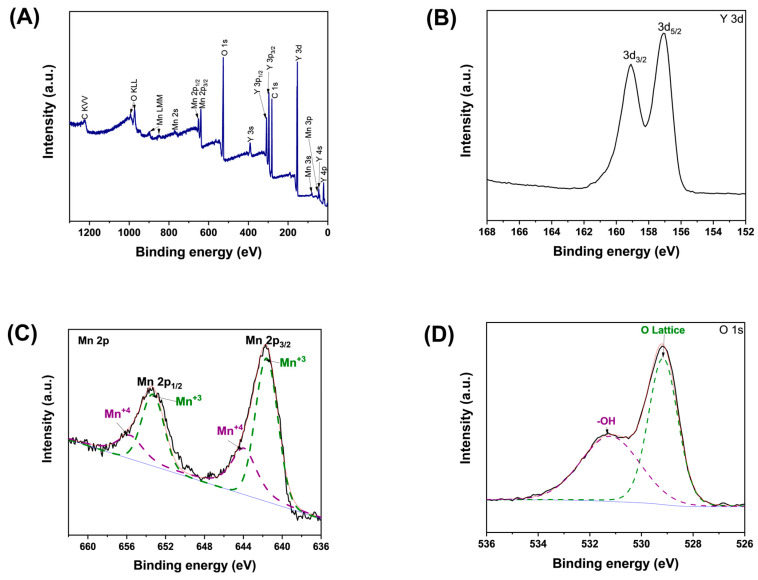
(**A**) Full XPS spectrum of the h-YMnO_3_ powders. Narrow scan of (**B**) Y 3d, (**C**) Mn 2p, and (**D**) O 1s levels.

**Figure 5 molecules-28-03932-f005:**
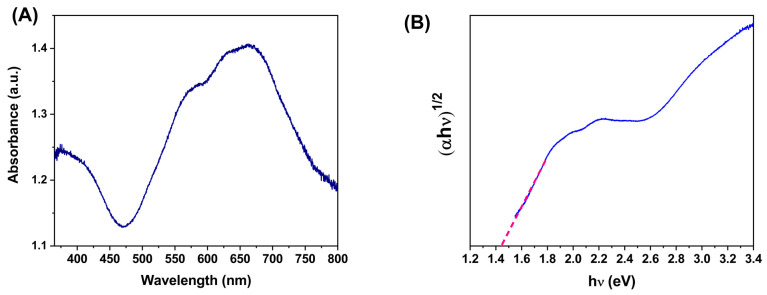
(**A**) UV-Vis absorbance spectrum of the h-YMnO_3_ powders. (**B**) Tauc plot was used to determine the band gap.

**Figure 6 molecules-28-03932-f006:**
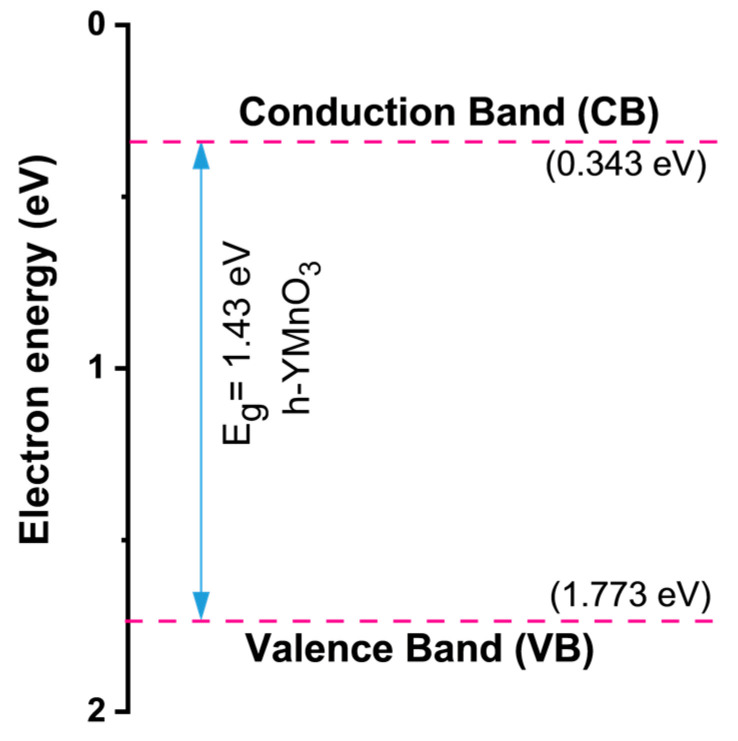
Conduction band (CB) and valence band (VB) positions for h-YMnO_3_.

**Figure 7 molecules-28-03932-f007:**
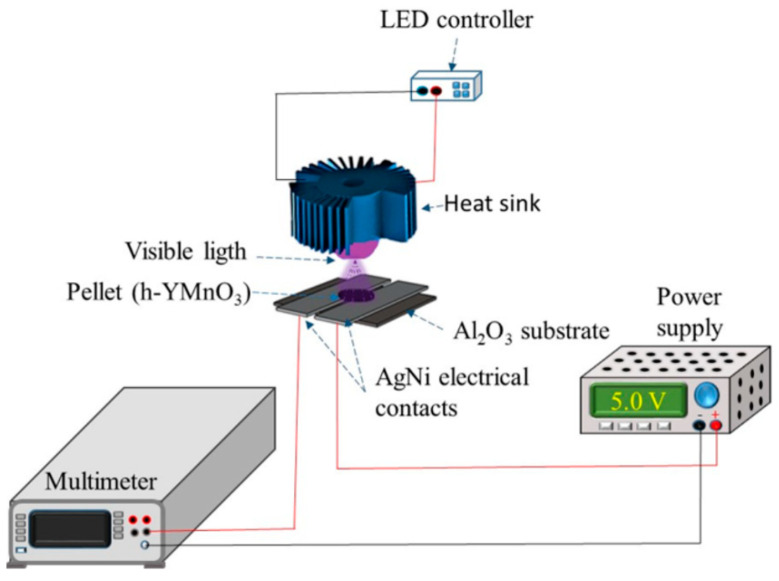
Diagram of the setup for the photocurrent test.

**Figure 8 molecules-28-03932-f008:**
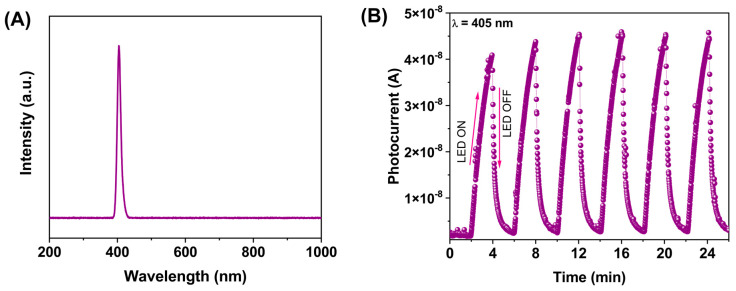
(**A**) Spectrum of the violet LED. (**B**) A plot of the photocurrent using violet light, with 2 min on/off cycles at the fixed optical power of 100 mW/cm^2^.

**Figure 9 molecules-28-03932-f009:**
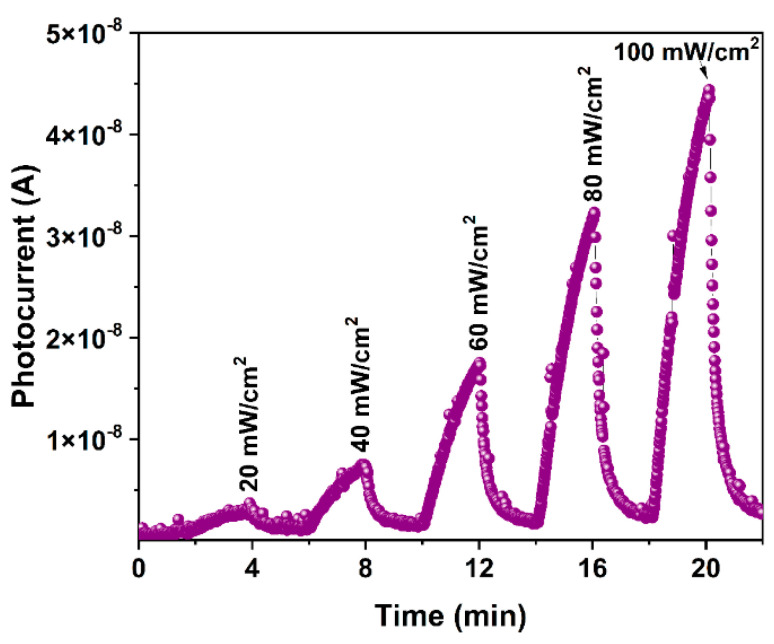
Photoelectric response at λ = 405 nm, using optical powers ranging from 20 to 100 mW/cm^2^, with on/off cycles of 2 min.

**Figure 10 molecules-28-03932-f010:**
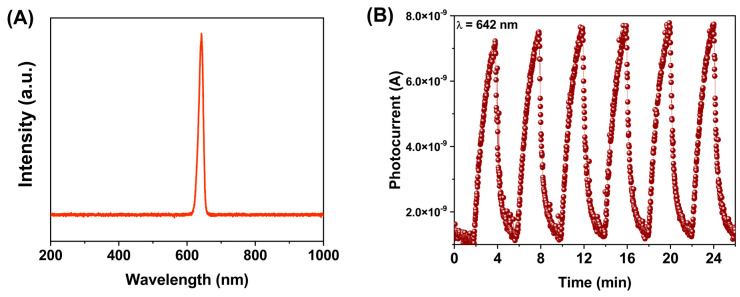
(**A**) Spectrum of the red LED. (**B**). A plot of the photocurrent using red light with 2 min on/off cycles at the fixed optical power of 100 mW/cm^2^.

**Figure 11 molecules-28-03932-f011:**
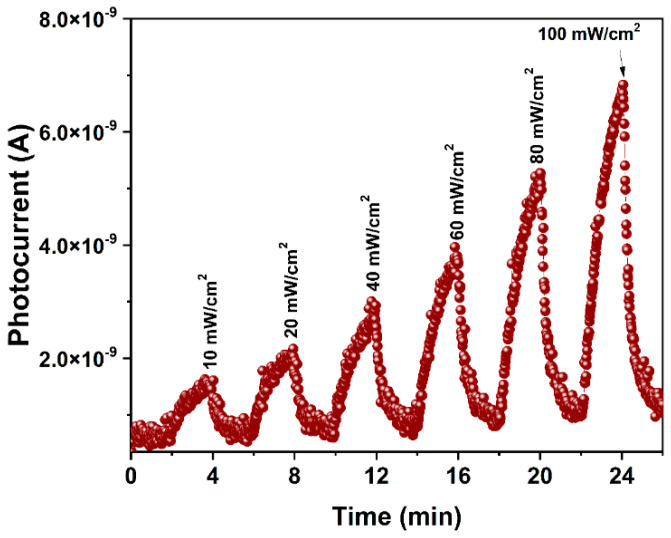
Photoelectric response at λ = 642 nm, using optical powers in the range of 10 to 100 mW/cm^2^, with on/off cycles of 2 min.

**Figure 12 molecules-28-03932-f012:**
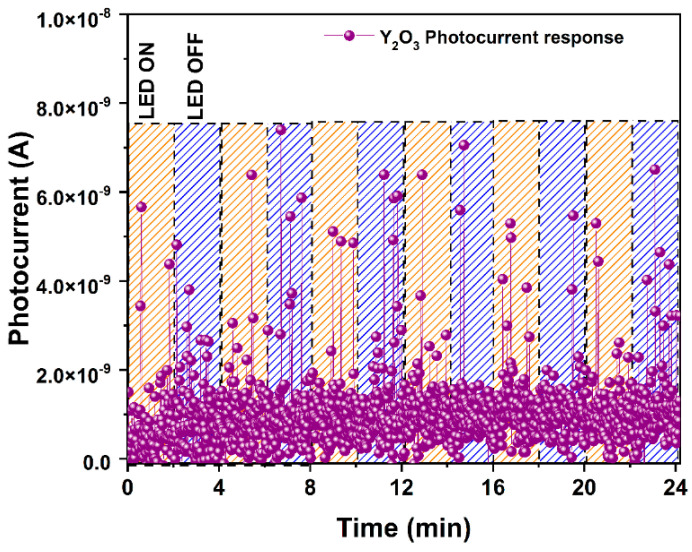
Photocurrent plot of Y_2_O_3_ pellet using violet light (λ = 405 nm), with 2 min on/off cycles at E_e_ = 100 mW/cm^2^.

**Figure 13 molecules-28-03932-f013:**
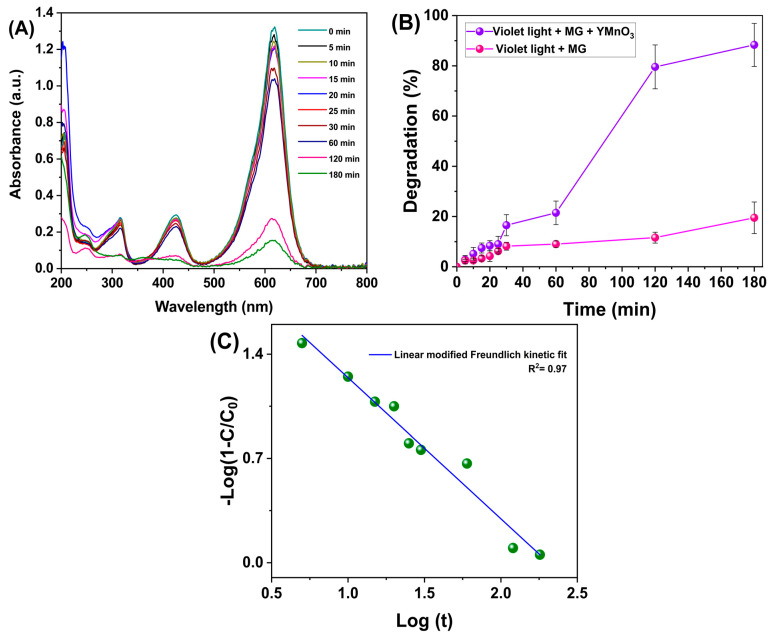
(**A**) UV-Vis absorbance spectrum of malachite green solutions containing h-YMnO_3_ as a photocatalyst after exposure to violet light. (**B**) Percentage degradation. (**C**) Linear fit corresponds to the modified Freundlich kinetic model.

**Figure 14 molecules-28-03932-f014:**
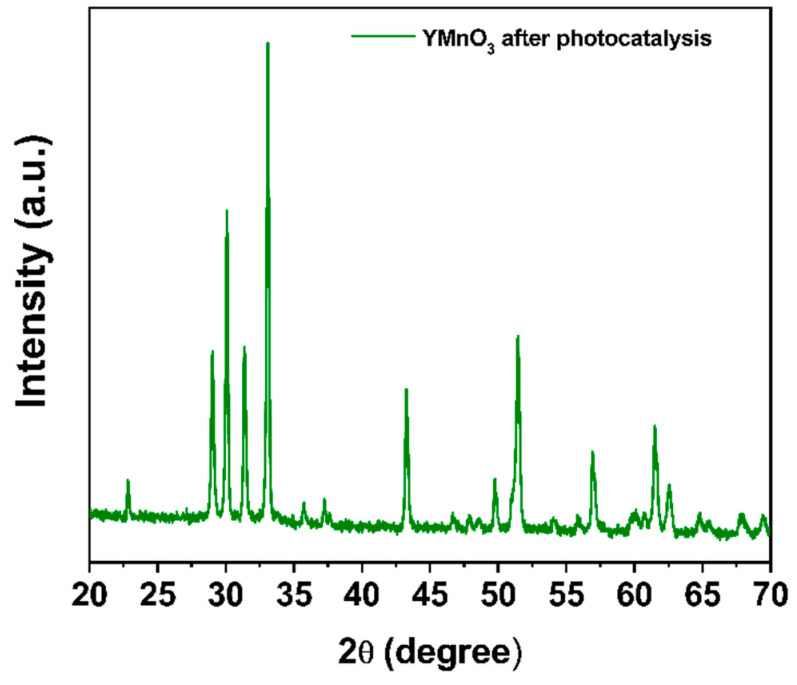
Diffraction pattern of the h-YMnO_3_ powders after being used as photocatalysts.

**Figure 15 molecules-28-03932-f015:**
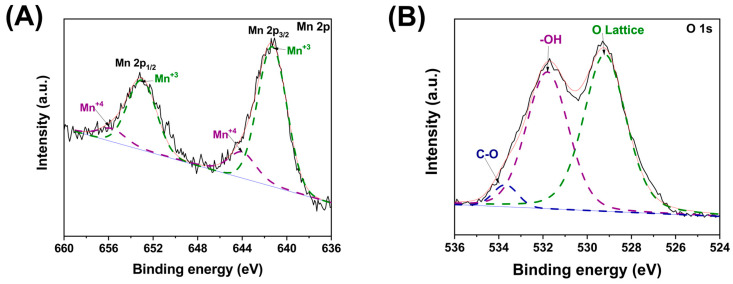
Narrow scan of Mn 2p and O 1s from the h-YMnO_3_ powders after being used as a photocatalysts. (**A**) Mn XPS spectrum; (**B**) O XPS spectrum.

**Figure 16 molecules-28-03932-f016:**
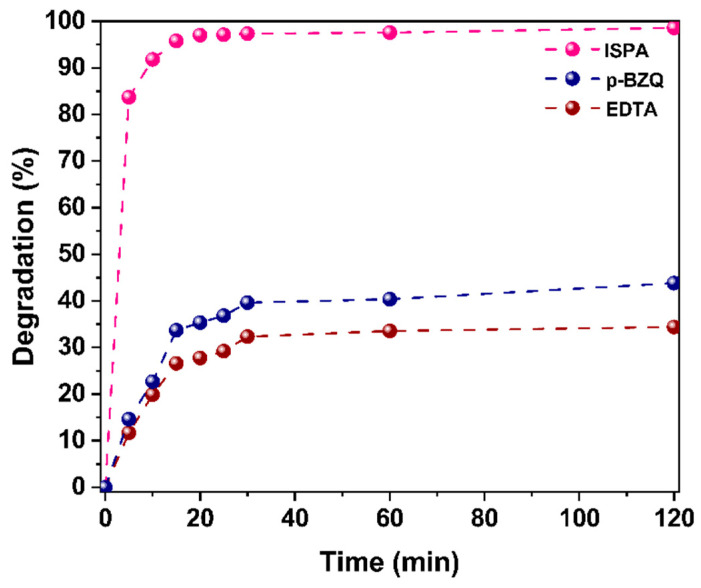
Effect of ISPA, p-BZQ, and EDTA scavengers on the photodegradation of the MG dye using h-YMnO_3_ as the photocatalyst and violet light with an *E_e_* = 100 mW/cm^2^.

**Figure 17 molecules-28-03932-f017:**
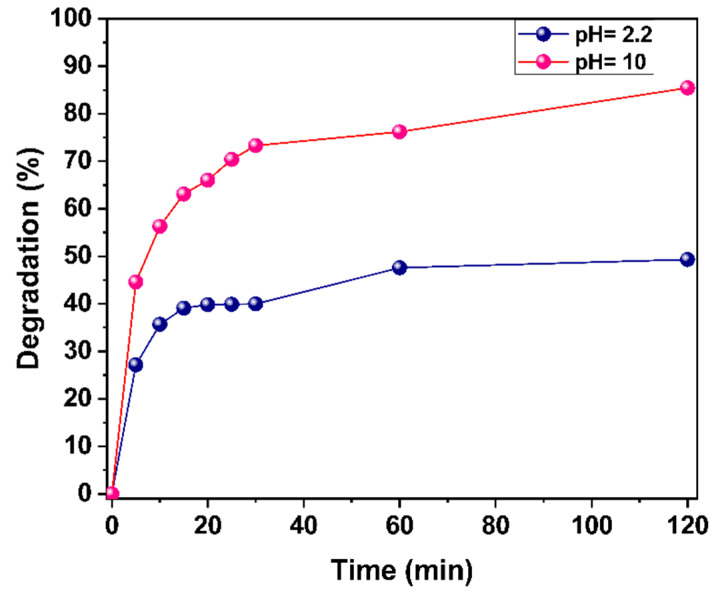
Contribution of pH to the photodegradation of the MG dye.

**Figure 18 molecules-28-03932-f018:**
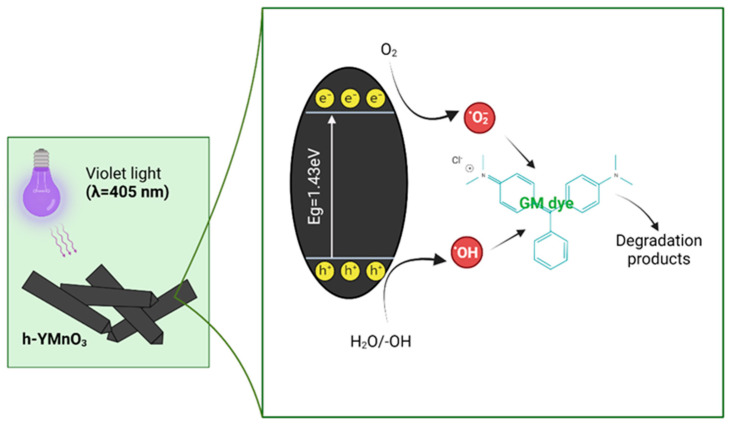
Photocatalytic degradation mechanism for the MG dye using h-YMnO_3_ as the photocatalyst under visible light exposure (λ = 405 nm).

**Figure 19 molecules-28-03932-f019:**
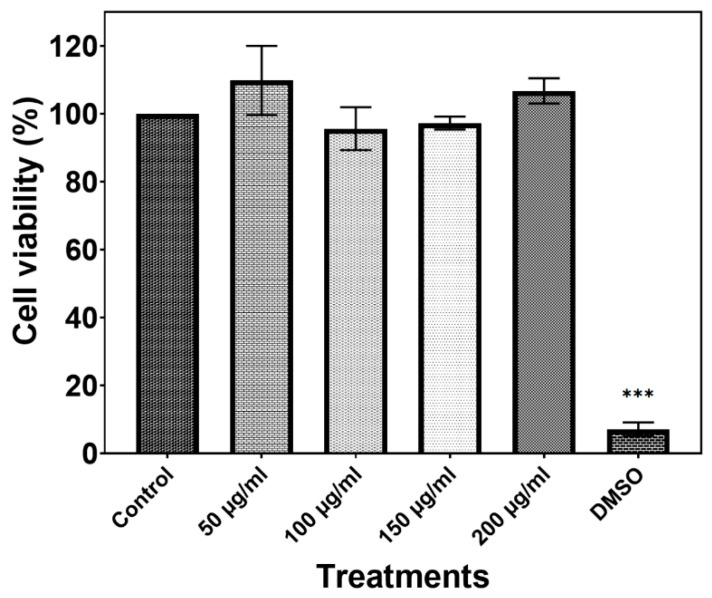
Bars represent at least three independent experiments and are plotted as mean ± SD. Significantly differences are depicted by *** indicate *p* < 0.0001.

## Data Availability

The data presented in this study are available upon request from the corresponding author.

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
