# Peer review of "Determining the Photoelectrical Behavior and Photocatalytic Activity of an h-YMnO3 New Type of Obelisk-like Perovskite in the Degradation of Malachite Green Dye"

_molecules, 2023, doi:10.3390/molecules28093932_

Round 1
Reviewer 1 Report
In this manuscript, the authors synthesized a rod-obelisk-shaped yttrium manganate perovskite (h-YMnO3) and investigated its photoelectrical behavior and photocatalytic activity in degradation of Malachite Green dye. It can be accepted after addressing the following concerns:
1. The electrochemical impedance spectroscopy should be supplemented to further study the photoelectrical behavior of h-YMnO3.
2. The band positions of h-YMnO3 should be analyzed.
3. In the Introduction section, some recent reports regarding perovskite-based photocatalysis should be cited, such as Chem. Eng. J. 2022, 433, 133762; Inorg. Chem. 2022, 61, 16028−16037; J. Mater. Chem. C, 2023, 11, 2540-2551.
Author Response
We thank you for the observations and we did our best to answer all your questions. In the attached file you will find these comments.

Reviewer 2 Report
In the present report, the authors constructed rod-obelisk-shaped yttrium manganate perovskite in its hexagonal phase through the precipitation method. Further, the photocatalytic activity has been tested for oxidative decomposition of malachite green under visible light, and Photoelectrical Behavior. This paper provided some valuable information and the content is very significant in this field. However, I recommended a major revision of the article from its present form before it can be published in molecules. Some specific comments are as follows:
1. The abstract and conclusion sections should be specific and scientific approach.
2. In the introduction section, the authors should expound on the research significance of the present work.
3. The authors should explain the novelty of the present report.
4. The authors should provide a schematic representation of the formation mechanism.
5. What is the pH of the reaction solution? The pH of the solution normally varies from precursor to precursor. The authors must justify the selection of pH, temperature, and time.
6. The authors should include error bars in all results for more accuracy.
7. What is the relationship between the optical bandgap and photocatalytic efficiency under light irradiation?
8. The abbreviations and labels are not properly organized throughout the manuscript.
9. The authors should provide TEM, high-resolution TEM, and lattice fringe patterns.
10. What is the key factor affecting photocatalytic efficiency?
11. Authors should perform scavenger tests, pH effects, loading effect, and photoluminescence analysis.
12. Authors should explain the mechanism with proper schematic representation.
13. All images are very poor resolution. Authors should produce high-quality images.
14. In the current state, there are more typographical errors and the language should be improved. Therefore, the authors are advised to recheck the whole manuscript for improving the language and structure carefully.
Author Response

(The authors gave the same response as above.)

Round 2
Reviewer 2 Report
The manuscript can be acceptable in the present form.